

# Impact of rainfall spatial aggregation on the identification of debris flow occurrence thresholds

Francesco Marra[1], Elisa Destro[2], Efthymios I. Nikolopoulos[3], Davide Zoccatelli[1,2], Jean Dominique Creutin[4], Fausto Guzzetti[5], and Marco Borga[2]

[1]Institute of Earth Sciences, Hebrew University of Jerusalem, Israel
[2]Department of Land, Environment, Agriculture and Forestry, University of Padova, Italy
[3]Department of Civil and Environmental Engineering, University of Connecticut, Storrs, USA
[4]Institut des Géosciences pour l'Environnement, Université de Grenoble Alpes/CNRS, France
[5]Istituto di Ricerca per la Protezione Idrogeologica, Consiglio Nazionale delle Ricerche, Perugia, Italy

*Correspondence to*: Francesco Marra (marra.francesco@mail.huji.ac.il)

**Abstract.** The systematic underestimation observed in debris flows early warning thresholds has been associated to the use of sparse rain gauge networks to represent highly non-stationary rainfall fields. Remote sensing products permit concurrent estimates of debris flow-triggering rainfall for areas poorly covered by rain gauges, but the impact of using coarse spatial resolutions to represent such rainfall fields is still to be assessed. This study uses fine resolution radar data for ~100 debris flows in the eastern Italian Alps to (i) quantify the effect of spatial aggregation (1–20-km grid size) on the estimation of debris flow triggering rainfall and on the identification of early warning thresholds and (ii) compare thresholds derived from aggregated estimates and rain gauge networks of different densities. The impact of spatial aggregation is influenced by the spatial organization of rainfall and by its dependence on the severity of the triggering rainfall. Thresholds from aggregated estimates show up to 8% and 21% variations in the shape and scale parameters respectively. Thresholds from synthetic rain gauge networks show >10% variation in the shape and >25% systematic underestimation in the scale parameter, even for densities as high as 1/10 km$^{-2}$.

## 1 Introduction

Debris flows are among the most impactful natural hazards in mountainous areas (Dowling and Santi, 2014; Badoux et al., 2016). They are primarily triggered by heavy rainfall hitting headwater catchments and, in many areas, their frequency is expected to increase in response to the intensification of the hydrological cycle caused by global warming (Westra et al., 2014; Dietrich and Krautblatter, 2017; Gariano and Guzzetti, 2016). Forecasting their occurrence is fundamental to save lives and properties and relies on early warning systems (Borga et al., 2014).

Operational debris flow early warning systems are largely based on empirical thresholds: meteorological and/or hydrological conditions above which debris flows are likely to occur (Caine, 1980). These thresholds are often defined as relationships between rain depth (or intensity) and duration, and are identified from past records (Guzzetti et al., 2008). Although rain gauge based thresholds are being used in different regions worldwide (Jakob et al., 2012; Segoni et al., 2015; Ma et al., 2015; Piciullo



et al., 2016), their accuracy depends on the abundance and quality of in-situ rainfall measurements and debris flow observations (Nikolopoulos et al., 2014).

The uncertainty related to rain gauge-based estimates of debris flow triggering rainfall is emphasized by the specific spatial organization of the triggering rainfall events. Recent studies, based on fine resolution radar estimates in the eastern Italian Alps, showed that the uncertainty related to rain gauge-based estimates of debris flow triggering rainfall is emphasized by the specific spatial organization of the triggering rainfall events: local rain depth peaks are often associated to the debris flow initiation points (Marra et al., 2016), and the occurrence and magnitude of such peaks depend on the severity of the triggering rainfall (Destro et al., 2017). This non-stationarity of the triggering rainfall fields causes systematic underestimation in the rain gauge estimates and propagates to the identification of the threshold relationships, decreasing their efficiency in separating triggering versus non-triggering events (Nikolopoulos et al., 2014, 2015a; Marra et al., 2014; Abancó et al., 2016).

Remote sensing rainfall products provide debris flow-concurrent estimates at the regional and global scales, allowing for the monitoring of areas poorly covered by rain gauge networks. Derivation of debris flow occurrence thresholds from such data sets can be used to develop local, regional or global warning systems. Moreover, it enables a direct and consistent comparison between thresholds in different regions, allowing to separate the contribution of hydro-geomorphological features other than rainfall, such as geology, pedology and slope, to the debris flows triggering. However, the use of remote sensing rainfall estimates is hampered by at least two factors: the coarse resolution of the products and the uncertainty due to current limitations in the retrieval algorithms. The vast majority of the studies dealing with remote sensing precipitation error over complex terrain focused on hydrological applications rather than debris flows or landslides triggering (Mei et al., 2014; Derin et al., 2016; Maggioni et al, 2017) or evaluated the combined effect of resolution and estimation uncertainty, without separating the contribution of each factor (Hong et al., 2006; Kirschbaum et al., 2012, 2013; Rossi et al., 2017; Nikolopoulos et al., 2017). Particularly in the case of debris flows, the small size of the initiation catchments is expected to highlight the impact of spatial resolution. It is thus important to quantify the effect of this factor on threshold based early warning systems.

In this study, we analyse the effect of using coarse resolution data for the identification of debris flows occurrence thresholds, leaving aside considerations on their prediction efficiency. In particular, (i) we quantify the impact of rainfall spatial aggregation, representing the resolution of remote sensed estimates, on the estimation error of debris flow triggering rainfall, and on the identification of the parameters of debris flows occurrence thresholds; and (ii) we compare thresholds derived from spatially aggregated rainfall estimates to those obtained from synthetic rain gauge networks of different densities, in order to assess the relative advantages of the two rainfall estimation methods. We explore the usual spatial scales of radar and satellite products (1–20-km grid size) and rain gauge network densities of $1/10$–$1/100$ km$^{-2}$.

## 2 Impact of spatial aggregation on the estimation of debris flow triggering rainfall

We build upon a unique high-resolution (1–km grid size, 5 min) data archive of radar rainfall estimates available for 11 storms which collectively triggered 99 debris flows in the eastern Italian Alps (Fig. 1). All the events are represented by channelized





debris flows triggered in very small basins. Event duration ranged between 1.5 and 26 h and the triggering rainfall between 8 and 180 mm. This data set is a representative sample (~20%) of the debris flows occurred in the study region in the period 2000-2014 (Nikolopoulos et al., 2015b). We refer to Destro et al. (2017) for additional information on the debris flows database and on the storm events. Weather radar rainfall estimates were corrected for errors due to attenuation in heavy rain, wet radome

attenuation, beam blockage, and vertical profile of reflectivity (Marra et al., 2014) and were then gauge-adjusted at the event scale using quality controlled rain gauge measurements. The radar data quality was checked at each step of the elaboration (Marra et al., 2014), and, as a result, the radar estimates are considered the best available spatial representation of debris flows triggering rainfall. To the authors' knowledge, this is among the largest and most accurate radar based dataset of debris flow triggering rainfall currently available worldwide.

Rainfall events were identified as separated by 24 h dry periods (Nikolopoulos et al., 2014). The triggering rain depth ($E$) was computed as the total rainfall estimated above the triggering location during the event duration ($D$). Spatially aggregated estimates were computed by spatially averaging the radar estimates on 1–20-km grid size. These scales reproduce the equivalent areal resolution of the most commonly used remote sensing rainfall products (Hsu et al., 1997; Hong et al., 2004; Huffman et al., 2007; Joyce et al., 2004; Huffman et al., 2015). To quantify the estimation error due to spatial aggregation, we

used the relative error ($RE$), calculated as:

$$RE = \frac{E}{E^*}, \tag{1}$$

where $E$ is the aggregated rainfall estimate and $E^*$ is the corresponding 1-km grid size value. Median and interquartile range (IQR) of the $RE$ were used to characterize the error distribution.

Following the results reported by Destro et al. (2017), we related the distribution of the estimation error to the severity of the

triggering rainfall. The return period of the triggering rainfall ($T$) was computed from depth-duration-frequency curves derived for the region using the method of the $L$-moments, and a kriging interpolation procedure in a regional Generalized Extreme Values framework (Destro et al., 2017). The severity of debris flow events was classified as mild ($T \leq$2y, 21 debris flows), moderate (2< $T \leq$50y, 41 debris flows) and severe ($T >$50y, 37 debris flows), depending on the return period of the 1-km grid size rainfall estimated above the triggering locations. Figure 2 shows the relative error for rainfall estimates aggregated on 1–

20-km grid sizes. Results are shown for all the debris flow events (Figure 2a) and for the three severity classes separately (Figure 2b).

When all the events are considered, the spatial aggregation yields a consistent decrease of the estimated rainfall, with $RE$ reaching 0.52 for 20-km grid size while the IQR increases up to ~0.4 for aggregation scales of ~13-km grid size. This suggests that an approximate 50% underestimation of the triggering rainfall amount is expected at 20-km grid size as a result of the

spatial aggregation alone. This pattern is caused by local rain depth peaks that correspond to the area most severely hit by the event-generating convective cells. Such peaks are frequently observed in close proximity of the triggering locations (Marra et al., 2016) so that, when rainfall is spatially aggregated, lower rainfall amounts are progressively included in the estimate.



Figure 1b shows that the scale-dependency pattern depends strongly on the severity of the triggering rainfall. Moderate and severe events (i.e., the 78 events triggered by $T>2y$ rainfall) exhibit similar underestimation, dominating the distribution observed in the general case. Conversely, mild events ($T≤2y$) show a substantially different pattern, with aggregated estimates overestimated even for 20-km grid sizes. In fact, as shown by Destro et al. (2017), debris flows triggered by short return period rainfall are often located close, but not corresponding, to the local rain depth peaks, so that the spatial aggregation is expected to include larger rain depths. The IQR is as high as 0.93 for 12-km grid size, and decreases to 0.8 for 20-km grid size.

## 3 Impact of spatial aggregation on the identification of debris flow occurrence thresholds

Rainfall thresholds for the occurrence of debris flows were derived in the form of power-law relationships between the triggering rain depth ($E$) and the rainfall duration ($D$), as in equation (2) (Guzzetti et al., 2008):

$$E = \alpha_p \cdot D^\beta \ . \tag{2}$$

The use of rainfall depth is equivalent to the use of rain intensity, but avoids the need to calculate rain intensity from the event rain depth and duration. The parameters of the threshold were objectively identified from the empirical data using the frequentist method proposed by Brunetti et al. (2010): the shape parameter $\beta$ was derived from the linear regression of the $(E, D)$ pairs plotted in logarithmic scale; the scale parameter $\alpha_p$ was calculated assuming a normal distribution of the regression residuals (log-residuals, hereinafter), and setting the required exceedance probability $p$ to the desired level. Therefore, the offset of the rainfall threshold with respect to the regression line depends on the distribution (i.e., standard deviation) of the log-residuals. In this study, we used $p = 5\%$, but the results we present hold for any probability level lower than 50%, i.e. for any threshold representing a lower envelop curve to the $(E, D)$ pairs. The debris flow occurrence thresholds were calculated, together with the regression relationships (i.e., the thresholds for $p = 50\%$) using the rain depth obtained from different aggregation scales, whereas the rainfall duration was kept unchanged. It can be pointed out that, in principle, a biased threshold may be efficient when used with equally biased rainfall estimates, however, Nikolopoulos et al. (2014, 2015a) showed that uncertainty in the estimation of the triggering rainfall strongly decreases the efficiency of thresholds in operational use.

Figure 3a-c shows the debris flow occurrence thresholds and regression relationships for rainfall aggregated at 1-, 10- and 20-km grid size. As shown by the slope of the relationships in Figure 3a-c, the shape parameter undergoes only minor variations (<8%) with spatial aggregation, implying that, in our sample, the distribution of event severity is approximately uniform with the event duration. Large differences are observed in the scale parameter of the regression relationships $\alpha_{50\%}$, which decreases from 26.5 at 1-km grid size to 13.4 at 20-km grid size. Conversely, minor variations are found for the scale parameter of the 5% threshold $\alpha_{5\%}$, which raises from 8.2 at 1-km grid size to 9.4 at 10-km grid size, and then decreases to 6.5 at 20-km grid size (Figure 3a-c). Figure 3d-f shows the distribution of the log-residuals as a function of the return period of the triggering rainfall for 1-, 10- and 20-km grid size. For 1-km grid size, the log residuals regularly scale with severity (Figure 3d). As the




information is spatially aggregated, this regular scaling is progressively lost (Figure 3e-f), due to the different effect of spatial aggregation on moderate/severe and mild events (Figure 2b).

## 4 Comparison between thresholds derived from spatially aggregated and rain gauge based estimates

Synthetic rain gauge networks were produced by randomly selecting the location of rain gauges to obtain densities of $1/A$,

with $A$ set to 10, 20, 50, and 100 km$^2$. To avoid clustering of the rain gauges, a minimum distance between two synthetic stations was set to $0.5\sqrt{A}$. Radar values of the pixels corresponding to the simulated gauge locations were used to mimic the synthetic rain gauge estimate. The rain gauge estimation of triggering rainfall was defined as the value reported by the rain gauge closest to the triggering location. This nearest neighbour method resulted as the less biased estimator for debris flow triggering rainfall, when compared to kriging and inverse squared distance methods (Nikolopoulos et al., 2015a; Destro et al.,

2017). The synthetic rain gauge estimation operation was iterated to obtain 100 Monte Carlo realizations for each rain gauge network density. Debris flow occurrence thresholds were derived for each realization and density and the estimated parameters were compared to the ones previously derived from spatially aggregated estimates. The uncertainty related to the synthetic networks parameters was quantified as the IQR of the corresponding set of realizations.

Figure 4 shows the relative error of the threshold and regression parameters derived from spatially aggregated (a) and rain-

gauge-based (b) estimates, together with the standard deviation of the respective log-residuals (c and d). As anticipated in the previous section, the shape parameter ($\boldsymbol{\beta}$) slightly increases with the aggregation scale, resulting in an 8% increase at 20-km grid size, whereas the regression scale parameter ($\boldsymbol{\alpha_{50\%}}$) exhibits a remarkable underestimation, with -51% at 20-km grid size (Figure 4a). The scale parameter of the threshold ($\boldsymbol{\alpha_{5\%}}$) increases up to ~7-km grid size ($RE$=1.19) and then monotonically decreases for larger grid sizes ($RE$=0.79 for 20-km grid size, Figure 4a). This dependence is explained by the combination of

two effects: (i) the mean value of the estimated rainfall decreases with spatial aggregation (Figure 2a); and (ii) the standard deviation of the log-residuals decreases by ~30% at 10-km grid size and remains almost unchanged for larger grid sizes (Figure 4c). This second effect is caused by the dependence of rainfall spatial pattern on the severity seen in Figure 2b: moderate/severe events are increasingly underestimated, while mild events are increasingly overestimated up to ~7-km grid size while, for larger grid sizes, the overestimation decreases.

Results obtained using synthetic rain gauge networks are remarkably different (Figure 4b). The shape parameter $\boldsymbol{\beta}$ exhibits a more marked increase and both $\boldsymbol{\alpha_{5\%}}$ and $\boldsymbol{\alpha_{50\%}}$ are systematically underestimated. Interestingly, the threshold scale parameter $\boldsymbol{\alpha_{5\%}}$ is more underestimated than the regression scale parameter $\boldsymbol{\alpha_{50\%}}$. This is explained by the large spatial variability of rainfall around the triggering locations: as the network density decreases, the estimation variance of synthetic rain gauge estimates increases causing the standard deviation of the log-residuals to increase (Figure 4d). As a result, debris flow

occurrence thresholds derived using very high density networks (1/10 km$^{-2}$) are comparable to the ones obtained using aggregation scales of 20-km grid size, corresponding to averaging areas as large as 400 km$^2$. Lower densities cause larger errors in both the shape parameter $\boldsymbol{\beta}$ and the scale parameter $\boldsymbol{\alpha_{5\%}}$ of the threshold.



In general, the estimation variance controls the scale parameter of the threshold relationship acting on the standard deviation of the log-residuals. In fact, an increased estimation variance increases the dispersion of the rainfall depth-duration pairs, causing the threshold relationship, a lower envelope curve, to decrease. As reported by Marra et al. (2016), even in presence of random estimation errors alone, an increase (decrease) of the estimation variance translates into an increase (decrease) of
the log-residuals, causing the threshold scale parameter to decrease (increase).

**5 Conclusions**

We examined the impact of using spatially aggregated estimates (1–20-km grid size) for the estimation of debris flow triggering rainfall and for the identification of debris flow occurrence thresholds in the form of power law depth-duration relationships. Results show that errors in the aggregated estimates of debris flow triggering rainfall strongly depend on the severity of the
triggering rainfall. Moderate to severe triggering rainfall (> 2y return period) is consistently underestimated, whereas mild triggering rainfall (≤ 2y return period) is generally overestimated up to 20-km grid size. The estimation variance controls the scale parameter of the threshold relationship, acting on the dispersion of the rainfall depth-duration pairs. The impact of spatial aggregation on the derivation of debris flow occurrence thresholds is thus controlled by the different response of mild and moderate/severe triggering rainfall to the spatial aggregation. Consequently, spatial aggregation causes up to 21% and 8%
variation in the scale and shape parameters of the identified debris flow occurrence thresholds. Conversely, rainfall estimates from synthetic rain gauge networks present large estimation variance so that debris flow occurrence thresholds derived from densities as high as 1/10 km$^{-2}$ are comparable to 20-km grid size spatial aggregation and thresholds from sparser networks are largely underestimated. These findings reveal a complex pattern of scale dependency, which should be considered when identifying and comparing triggering rainfall amounts or debris flow occurrence thresholds derived from different products.
In general, rainfall spatial aggregation is more consistent than rain gauge based estimation in the identification of the threshold parameters across the examined spatial scales.

Results from this study are subject to uncertainty due to the limited size of the dataset, however, to the authors' knowledge, this is among the largest and most accurate dataset of debris flow triggering rainfall currently available worldwide. It includes ~20% of the debris flows occurred in the study region in the last 15 years and can thus be considered quantitatively
representative of the climatology of debris flow triggering rainfall in the area. Given the general and objective formulation used, results from this study are qualitatively transferrable to those situations in which lower envelop curves are used to predict the occurrence of point-like events in presence of non-stationary fields, such as in the case of shallow landslides (Guzzetti et al., 2008) or urban flooding (Yang et al., 2016) prediction.

Future studies, quantifying the impact of (i) estimation uncertainty due to rainfall retrieval algorithms and (ii) temporal
resolution of remote sensing data, are required before the operational use of remote sensed rainfall products in debris flow early warning systems. To this end, the performance of new satellite-rainfall products (e.g. IMERG Huffman et al. 2015), available at high space/time resolution (e.g. 0.1deg/0.5h), deems promising and calls for subsequent analysis.



*Data availability.* Weather radar and debris flow data used for this study are available in the HYLAND project website (http://intra.tesaf.unipd.it/cms/hyland/).

*Competing interests.* The authors declare no competing interest.

*Acknowledgements.* Marra was supported by the Lady Davis Fellowship Trust [Project: RainFreq]. We acknowledge the daily efforts of Ripartizione Opere Idrauliche, Autonomous Province of Bolzano (Italy), for updating the debris flows database and Ufficio Idrografico, Autonomous Province of Bolzano (Italy), Mauro Tollardo in particular, for providing access to the radar data.

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




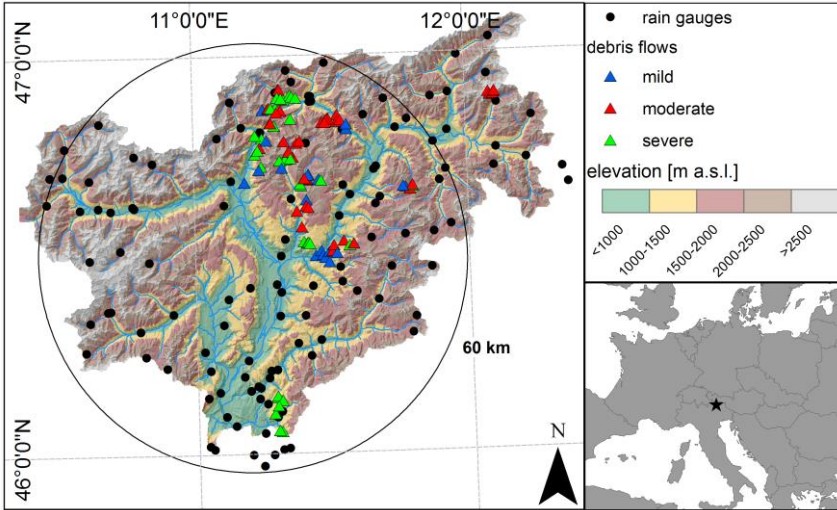

**Figure 1: Map of the study area including terrain elevation, rain gauge network (dots) and debris flows included in the study (triangles). Location of the study area within Europe is shown.**



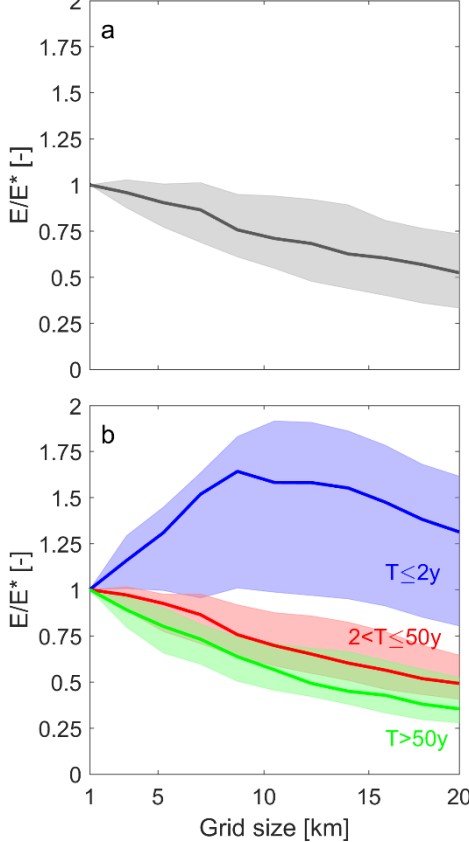

**Figure 2: Relative error in the rainfall estimates as a function of the aggregation scale for (a) all the 99 considered debris flow events, and (b) different classes of severity (return period of the 1-km gird size triggering rainfall). Solid lines show median values and shaded areas the interquartile ranges.**



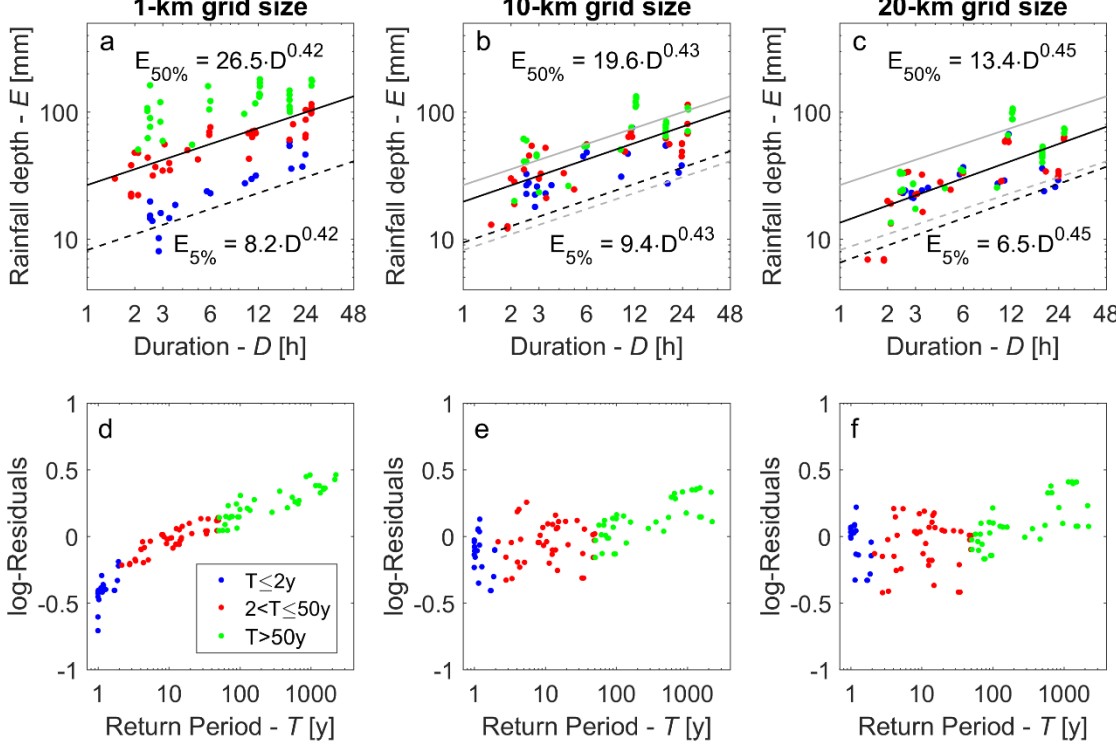

**Figure 3: Threshold and regression relationships for 1-, 10- and 20-km grid sizes (a-c, respectively) and corresponding residuals of the regressions in logarithmic scale (log-residuals) as a function of the return period of the 1-km triggering rainfall (d-f). Light grey lines in (b, c) reproduce, for reference, the 1-km grid size relationships.**



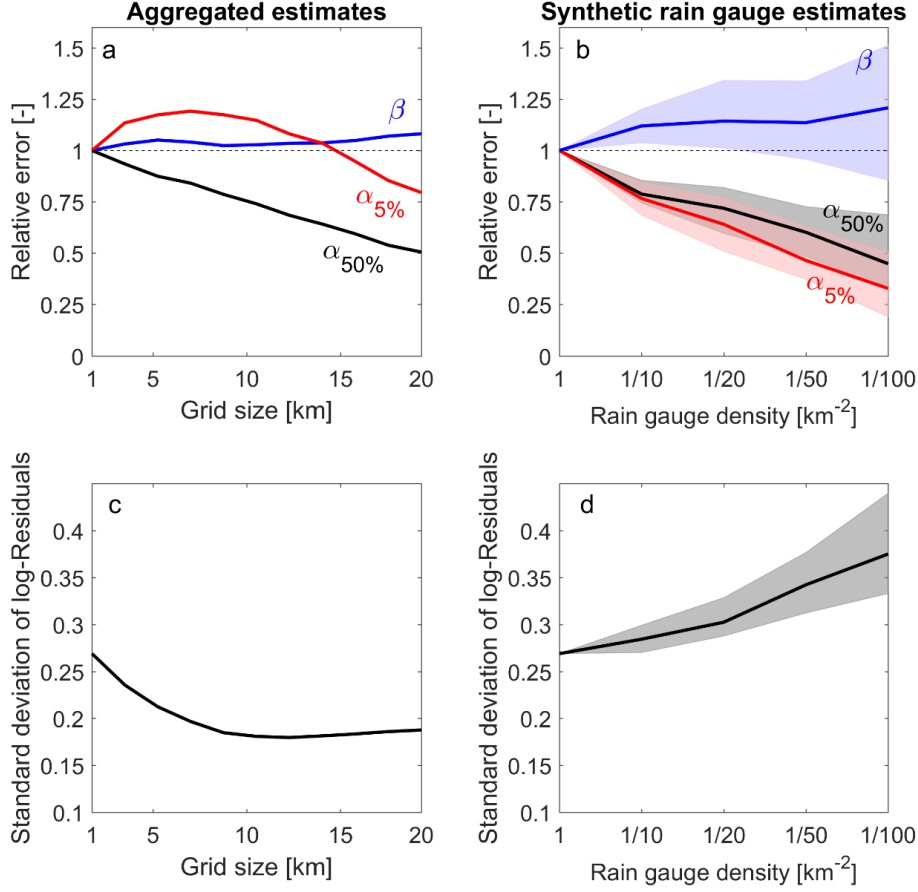

**Figure 4: Relative error of the parameters of the debris flow occurrence thresholds and associated regressions (a-b) and standard deviation of the log-residuals (c-d) obtained from spatial aggregation of rainfall (a, c), and nearest neighbour interpolation of synthetic rain gauge networks (b, d) – shaded areas represent the interquartile range of the Monte Carlo realizations.**