# Peer review of "Impact of rainfall spatial aggregation on the identification of debris flow occurrence thresholds"

_Hydrology and Earth System Sciences, 2017_

## Referee Comment (RC1) · Anonymous Referee #1 · 17 Jul 2017

**General comments**

Consistently with its title, the paper analyses how the spatial scale of aggregation of rainfall can influence the determination of intensity-duration debris flow occurrence thresholds. It distinguishes between the two cases of (i) regular grids, and (hypothetical) (ii) rain gauge networks. It capitalizes on a data set of 1 km/5 min radar rainfall and 99 debris flow events. The paper fits within the scope of HESS, it is well written, scientific questions are clear and relevant, and conclusions are supported by the results. Nevertheless, the paper may benefit from a more in-depth analysis regarding the available methods for threshold determination, as I describe in the first point of the "specific comments". For this reason, I suggest moderate revisions for the manuscript to be finally published in HESS.

[Figure]

Specific comments

Section 3: The so-called frequentist method for threshold determination originally proposed by Brunetti et al. (2010), involves only triggering events. Relatively recent research has highlighted the importance to take into account also non-triggering events. To consider only triggering events generally brings to thresholds that are lower than many non-triggering events, and thus a high number of false alarms, which may generate a disbelief in the early warning system (e.g. Berti et al., 2013 doi:10.1029/2012JF002367). The authors should at least should discuss the drawbacks of the method proposed by Brunetti et al. (2010). Possibly, the authors should add to the paper the same analysis they have conducted, but for the case that threshold determination is conducted by taking into account both triggering and non-triggering events.

P5 L4: Few details are given about the method for generating the synthetic rainfall fields. Since the method may affect the results, please provide these details.

P3 L23-24: "The severity of debris flow events was classified as mild (T<=2y, 21 debris flows), moderate (2<T< 50y, 41 debris flows) ... ". The authors should mention that return period of a debris flow depends in general from both initial conditions and triggering rainfall, and not only from the latter (Peres and Cancelliere, 2016; http://dx.doi.org/10.1016/j.jhydrol.2016.03.036). Furthermore, intra-event rainfall intensity variability, which also affects return period, may not be properly taken into account with depth-duration-frequency curves (see D'Odorico et al., 2005; doi:10.1029/2004JF000127).

Technical corrections

P2 L8. Perhaps "variability" is more appropriate than "non-stationary"

---

## Referee Comment (RC2) · Anonymous Referee #2 · 18 Jul 2017

General Comments

In this manuscript the authors investigate the effects of spatial aggregation of precipitation on the power law Total Raingall-Duration thresholds for debris flow. The study is based on 11 storms inducing 99 debris flow events in a region in the North of Italy and uses 5-min radar data with a spatial resolution of 1 km2. The spatial aggregation ranges from 1 to 20 km cells, corresponding to resolutions typical of remote sensing data. Additionally the authors compare the results of the spatial aggregation with those obtained with a synthetic raingauge network of different densities (1/10 up to 1/100 km2). Overall the paper is well written, with a clear structure and objective. I believe it could benefit from some more elaborations on some of the aspects presented, mentioned here below. I recommend minor revisions before publication on the journal.

[Figure]

Specific Comments

P5 L27-31: The authors conclude from Figure 4 and the increase of the log-residuals' standard deviation with decreasing synthetic raingauges network density that "debris flow occurrence thresholds derived using very high density networks (1/10 km-2) are comparable to the ones obtained using aggregation scales of 20-km grid size, corresponding to averaging areas as large as 400 km2". This conclusions seems to be based on the relative error of the alpha parameter with p=5% (ca. 25% underestimation for both 20-km grid size and 1/10 km2 network density) and therefore be true only for this specific method chosen (following Brunetti et al., 2010). For instance the alpha for p=50% for 20km grid size corresponds roughly to 1/100km2 density.

P6 Conclusions: on the same line of the comment above, the authors could elaborate a bit more on the effect of the specific choice of method made and its effects on the conclusions. For instance, how do the authors believe the results would changing when applying a different method for the definition of the thresholds? Already the choice of p seems to affect the conclusions. Furthermore, how would the results change when applying a method that accounts not only for triggering events, but also non-triggering events?

Technical Corrections

P4 L1: should be Figure 2b

---

## Referee Comment (RC3) · Anonymous Referee #3 · 20 Jul 2017

The manuscript of Marra et al. seems to me as a very good and interesting piece of research on the rainfall trigger conditions for debris flows. The paper is well-written, the structure is appropriate. Some minor comments: P3, L11: maybe clarify how the end of the rainfall event was defined. Is it the time of DF initiation or end of rainfall? Was the definition of the rainfall event for the rain gauge station data always unambiguous? P4, L1: should be Figure 2b. P5, L4: I agree with referee1 that a few more details on the method for generating the synthetic rainfall fields would be useful.

---

## Author Comment (AC1) · 25 Jul 2017

**Manuscript reference number: hess-2017-308 - Response to Anonymous referee #1**

We would like to thank the referee for his review and for the helpful comments. We provide here a response to his comments together with our proposed edits to the manuscript. The referee's comments are reported in black and denoted as RXCY where X is the reviewer number and Y is the corresponding comment number whereas our response is in blue.

Consistently with its title, the paper analyses how the spatial scale of aggregation of rainfall can influence the determination of intensity-duration debris flow occurrence thresholds. It distinguishes between the two cases of (i) regular grids, and (hypothetical) (ii) rain gauge networks. It capitalizes on a data set of 1 km/5 min radar rainfall and 99 debris flow events. The paper fits within the scope of HESS, it is well written, scientific questions are clear and relevant, and conclusions are supported by the results. Nevertheless, the paper may benefit from a more in-depth analysis regarding the available methods for threshold determination, as I describe in the first point of the "specific comments". For this reason, I suggest moderate revisions for the manuscript to be finally published in HESS.

We would like to thank the referee for his review.

R1C1

Section 3: The so-called frequentist method for threshold determination originally proposed by Brunetti et al. (2010), involves only triggering events. Relatively recent research has highlighted the importance to take into account also non-triggering events. To consider only triggering events generally brings to thresholds that are lower than many non-triggering events, and thus a high number of false alarms, which may generate a disbelief in the early warning system (e.g. Berti et al., 2013 doi:10.1029/2012JF002367). The authors should at least should discuss the drawbacks of the method proposed by Brunetti et al. (2010). Possibly, the authors should add to the paper the same analysis they have conducted, but for the case that threshold determination is conducted by taking into account both triggering and non-triggering events.

Thank you for bringing up this point. Oftentimes, the frequentist method is thought to have issues with false alarms. However, in our view, we should more appropriately talk about positives/negatives rather than 'alarms'. In fact, going from 'thresholds' to 'alarm' requires the definition of rules, that represent a complicated and often overlooked issue (see e.g., Piciullo et al. 2017, doi:10.1007/s10346-016-0750-2, and references therein). In addition, unfortunately, an unknown number of so called 'non-triggering' events may actually have caused landslides that simply went unnoticed or unreported (see e.g., Gariano et al. 2015, doi:10.1016/j.geomorph.2014.10.019). This is a well-known problem for non-instrumental measures such as 'landslide' vs. 'no landslide'. For this reason, using 'non-triggering' events may result in too high thresholds and consequently in an increased number of false negatives which, in an alarm system, are generally much worse than false positives.

In addition, in operational environment, the spatial-temporal characteristics of rainfall are rarely considered. Conversely, our research (here and in Marra et al. 2016, doi:10.1016/j.jhydrol.2015.10.010) confirms the presence of (spatial) non-stationarity of the rainfall fields around the debris flow triggering locations, meaning that the spatial organization characteristics of triggering and non-triggering rainfall are completely different (see Fig. R.1). It would be thus misleading to include such different patterns in the analysis.

To conclude, given the generality of the findings ("qualitatively transferrable to those situations in which lower envelop curves are used to predict the occurrence of point-like events in presence of non-stationary fields" [P6L26]), we think that using an (i) objective and (ii) straightforward method, such as the frequentist, is crucial to allow a clear interpretation of the study.

As suggested by the referee we think it is worth discussing this choice in the updated manuscript: "*In operational environment, the spatial-temporal characteristics of rainfall have rarely been considered,*

*despite the observed non-stationarity of the rainfall fields around the debris flow triggering locations reported by Marra et al. (2016). Since our objective is to analyse the impact of this non-stationarity on the use of spatially aggregated rainfall information, it is crucial for us to focus on the triggering events, i.e. on the events in which the systematic spatial feature is observed, and to use an objective and straightforward method, such as the frequentist, that allows a clear interpretation of the results*".

[Figure]

Figure R.1: mean normalized rainfall fields (i.e. the central point is 1) calculated for the events included in Marra et al. (2016) for (a) debris flow locations ('triggering') and (b) random points within the radar field ('non-triggering'). The non-stationarity of the triggering rainfall fields is clear. Note that (a) is analogous to Fig. 6 in Marra et al. (2016) with different color scale (here is log-scaled) and color bar

R1C2
P5 L4: Few details are given about the method for generating the synthetic rainfall fields. Since the method may affect the results, please provide these details.
Thank you for the question. The method is based on generating rain gauge networks (i.e. coordinates of hypothetical rain gauges) rather than rainfall fields. The synthetic rain gauge estimates are defined as the radar measurements on the corresponding pixels (i.e. the pixels containing the location of the rain gauge). The approach of using the radar rainfall fields as the 'true' rainfall fields follows exactly what was done for the analysis of spatial aggregation. The triggering rainfall is then defined by the measurement of the rain gauge closest to each debris flow (nearest neighbor 'interpolation' method). This approach strictly follows what previously used by Nikolopoulos et al. (2015, doi:10.5194/nhess-15-647-2015) and Destro et al. (2017, doi:10.1016/j.geomorph.2016.11.019). We propose to update this portion of the manuscript to improve its clarity: *"Synthetic rain gauge networks were produced using the procedure proposed by Nikolopoulos et al. (2015a) and Destro et al. (2017). The location of the rain gauges was randomly generated to obtain densities of $1/A$, with $A$ set to 10, 20, 50, and 100 $km^2$. To avoid clustering of the rain gauges, a minimum distance between two synthetic stations was set to $0.5\sqrt{A}$. Rainfall estimates of the synthetic rain gauges were defined as the value of the radar rainfall fields for the pixels corresponding to the simulated gauge locations. The rain gauge estimation of triggering rainfall was then defined as the value reported by the rain gauge closest to the triggering location."*

R1C3
P3 L23-24: "The severity of debris flow events was classified as mild (T<=2y, 21 debris flows), moderate (2<T< 50y, 41 debris flows)...". The authors should mention that return period of a debris flow depends

in general from both initial conditions and triggering rainfall, and not only from the latter (Peres and Cancelliere 2016, http://dx.doi.org/10.1016/j.jhydrol.2016.03.036). Furthermore, intra-event rainfall intensity variability, which also affects return period, may not be properly taken into account with depth-duration-frequency curves (see D'Odorico et al. 2005, doi:10.1029/2004JF000127).

Thank you for pointing this out. The use of 'severity of debris flow events' is actually misleading, since we only refer to the return period of the triggering rainfall and do not include other sources of information. We propose to update this part of the text to: "*The severity of the debris-flow triggering rainfall was classified as…*".

Technical corrections

R1C4

P2 L8. Perhaps "variability" is more appropriate than "non-stationary"

We respectfully disagree with the reviewer on this suggestion. What is meaningful reporting from Marra et al. (2016) is the presence of systematic patterns in the triggering rainfall fields around the debris flows, meaning a non-stationarity, observed at least for the first moment. The use of 'variability' is not sufficient to explain these findings and, in fact, this observed non-stationarity is among the motivations behind this study (see the abstract: "The systematic underestimation observed in debris flows early warning thresholds has been associated to the use of sparse rain gauge networks to represent highly non-stationary rainfall fields" [P1L11]). Please, see also Fig. R.1 and the related text in the response to R1C1 for more details.

---

## Author Comment (AC2) · 25 Jul 2017

**Manuscript reference number: hess-2017-308 - Response to Anonymous referee #2**

We would like to thank the referee for his review and for the interesting comments. We provide here a response to his comments together with our proposed edits to the manuscript. The referee's comments are reported in black and denoted as RXCY where X is the reviewer number and Y is the corresponding comment number whereas our response is in blue.

In this manuscript the authors investigate the effects of spatial aggregation of precipitation on the power law Total Raingall-Duration thresholds for debris flow. The study is based on 11 storms inducing 99 debris flow events in a region in the North of Italy and uses 5-min radar data with a spatial resolution of 1 km2. The spatial aggregation ranges from 1 to 20 km cells, corresponding to resolutions typical of remote sensing data. Additionally the authors compare the results of the spatial aggregation with those obtained with a synthetic raingauge network of different densities (1/10 up to 1/100 km2). Overall the paper is well written, with a clear structure and objective. I believe it could benefit from some more elaborations on some of the aspects presented, mentioned here below. I recommend minor revisions before publication on the journal.

We would like to thank the referee for his review.

R2C1

P5 L27-31: The authors conclude from Figure 4 and the increase of the log-residuals' standard deviation with decreasing synthetic raingauges network density that "debris flow occurrence thresholds derived using very high density networks (1/10 km-2) are comparable to the ones obtained using aggregation scales of 20-km grid size, corresponding to averaging areas as large as 400 km2". This conclusions seems to be based on the relative error of the alpha parameter with p=5% (ca. 25% underestimation for both 20-km grid size and 1/10 km2 network density) and therefore be true only for this specific method chosen (following Brunetti et al., 2010). For instance the alpha for p=50% for 20km grid size corresponds roughly to 1/100km2 density.

Thank you for pointing out this aspect. We agree with the referee on this comment and we would like to update the text in the manuscript to highlight this aspect: "*As a result, debris flow occurrence thresholds obtained using aggregation scales of 20-km grid size (corresponding to averaging areas as large as 400 km2) are comparable to the ones derived from relatively high-density rain gauge networks, such as 1/10 km-2 for 5% exceedance probability thresholds or 1/100 km-2 for 50% exceedance probability thresholds.*" Moreover, a sentence in the conclusions would be updated accordingly (see response to R2C2).

R2C2

P6 Conclusions: on the same line of the comment above, the authors could elaborate a bit more on the effect of the specific choice of method made and its effects on the conclusions. For instance, how do the authors believe the results would changing when applying a different method for the definition of the thresholds? Already the choice of p seems to affect the conclusions.

As underlined in the response to R2C1, the choice of the exceedance probability level used for the thresholds does actually affect the quantitative values reported in the above mentioned comment and a sentence in the conclusions should be updated to decrease the impact of the reported example: "*Conversely, rainfall estimates from synthetic rain gauge networks present large estimation variance so that debris flow occurrence thresholds at 5% exceedance probability derived from densities as high as 1/10 km$^{-2}$ are can be comparable to 20-km grid size spatial aggregation and thresholds from sparser networks are largely underestimated*"

However, this does not affect the conclusions of the study that are pointed towards the use thresholds as "lower envelop curves" [P4L18; P6L3; P6L23] and are not specific (except the particular example

mentioned above) of any particular choice of exceedance probability (in the case of frequentist) or of the triggering events-based method used, as soon as an exceedance probability lower than 50% is used ("the results we present hold for any probability level lower than 50%, i.e. for any threshold representing a lower envelop curve to the (E, D) pairs" [P4L17-18]).

R2C3
Furthermore, how would the results change when applying a method that accounts not only for triggering events, but also non-triggering events?
The use of non-triggering events for the definition of the thresholds has been thoroughly discussed in our response to referee #1. We would like here to refer, in particular, to our response to R1C1.

R2C4
P4 L1: should be Figure 2b
Thank for noticing. The figure reference will be updated.

---

## Author Comment (AC3) · 25 Jul 2017

**Manuscript reference number: hess-2017-308 - Response to Anonymous referee #3**

We would like to thank the referee for his review. We provide here a response to his comments together with our proposed edits to the manuscript. The referee's comments are reported in black and denoted as RXCY where X is the reviewer number and Y is the corresponding comment number whereas our response is in blue.

The manuscript of Marra et al. seems to me as a very good and interesting piece of research on the rainfall trigger conditions for debris flows. The paper is well-written, the structure is appropriate.
We would like to thank the referee for his review.

R3C1
P3, L11: maybe clarify how the end of the rainfall event was defined. Is it the time of DF initiation or end of rainfall? Was the definition of the rainfall event for the rain gauge station data always unambiguous?
This is a good question. Unfortunately, the temporal information on the DF occurrence in the catalog is often as high as the day of occurrence, so that no information is available on the time of occurrence. Consequently, the rainfall events are identified until the end of the rainfall. The definition was not ambiguous since each DF was associated to a single rain gauge (nearest neighbor approach), thus providing single rainfall data series.
We propose to update this part of the text to make this aspect clearer to the readers adding: "*Since no information is available on the exact time of occurrence of the debris flows, the events were extended until the end of the rainfall.*"

R3C2
P4, L1: should be Figure 2b.
Thanks for noticing. The figure reference will be updated.

R3C3
P5, L4: I agree with referee1 that a few more details on the method for generating the synthetic rainfall fields would be useful.
Thank you for the question. As reported in the response to referee #1: the method is generating rain gauge networks (i.e. coordinates of hypothetical rain gauges) rather than rainfall fields. The synthetic rain gauge estimates are defined as the radar measurements on the corresponding pixels (i.e. the pixels containing the location of the rain gauge). The approach of using the radar rainfall fields as the 'true' rainfall fields follows exactly what was done for the analysis of spatial aggregation. The triggering rainfall is then defined by the measurement of the rain gauge closest to each debris flow (nearest neighbor 'interpolation' method). This approach strictly follows what previously used by Nikolopoulos et al. (2015) and Destro et al. (2017). We propose to update this portion of the manuscript to improve its clarity: "*Synthetic rain gauge networks were produced using the procedure proposed by Nikolopoulos et al. (2015a) and Destro et al. (2017). The location of the rain gauges was randomly generated to obtain densities of $1/A$, with $A$ set to 10, 20, 50, and 100 $km^2$. To avoid clustering of the rain gauges, a minimum distance between two synthetic stations was set to $0.5\sqrt{A}$. Rainfall estimates of the synthetic rain gauges were defined as the value of the radar rainfall fields for the pixels corresponding to the simulated gauge locations. The rain gauge estimation of triggering rainfall was then defined as the value reported by the rain gauge closest to the triggering location.*"

---

## Author Response (AR2)

Dear editor,

Thank you for your suggestion concerning the abstract. We modified the last sentences to "*Thresholds from aggregated estimates show up to 8%–21% variations in the parameters whereas more than 10%–25% systematic variations result from the use of rain gauge networks, even for densities as high as 1/10 km$^{-2}$.*" In order to have it less specific and more appealing to the reader.

Kind regards,
Francesco Marra